# New AI-algorithms on smartphones to detect skin cancer in a clinical setting—A validation study

Teresa Kränke[1]*, Katharina Tripolt-Droschl[1], Lukas Röd[2], Rainer Hofmann-Wellenhof[1], Michael Koppitz[2], Michael Tripolt[1]

**1** Department of Dermatology and Venereology, Medical University of Graz, Graz, Austria, **2** Medical University of Graz, Graz, Austria

* teresa.kraenke@medunigraz.at

## Abstract

### Background and objectives

The incidence of skin cancer is rising worldwide and there is medical need to optimize its early detection. This study was conducted to determine the diagnostic and risk-assessment accuracy of two new diagnosis-based neural networks (analyze and detect), which comply with the CE-criteria, in evaluating the malignant potential of various skin lesions on a smartphone. Of note, the intention of our study was to evaluate the performance of these medical products in a clinical setting for the first time.

### Methods

This was a prospective, single-center clinical study at one tertiary referral center in Graz, Austria. Patients, who were either scheduled for preventive skin examination or removal of at least one skin lesion were eligible for participation. Patients were assessed by at least two dermatologists and by the integrated algorithms on different mobile phones. The lesions to be recorded were randomly selected by the dermatologists. The diagnosis of the algorithm was stated as correct if it matched the diagnosis of the two dermatologists or the histology (if available). The histology was the reference standard, however, if both clinicians considered a lesion as being benign no histology was performed and the dermatologists were stated as reference standard.

### Results

A total of 238 patients with 1171 lesions (86 female; 36.13%) with an average age of 66.19 (SD = 17.05) was included. Sensitivity and specificity of the detect algorithm were 96.4% (CI 93.94–98.85) and 94.85% (CI 92.46–97.23); for the analyze algorithm a sensitivity of 95.35% (CI 93.45–97.25) and a specificity of 90.32% (CI 88.1–92.54) were achieved.

### Discussion

The studied neural networks succeeded analyzing the risk of skin lesions with a high diagnostic accuracy showing that they are sufficient tools in calculating the probability of a skin

**Data Availability Statement:** All relevant data are within the paper and its Supporting information files.

**Funding:** The development of the algorithms was funded by the amiflow Ltd., Graz, Austria; however, the authors did not receive any salaries by this company nor did it have any influence on the study design, data collection and analysis, decision to publish, or preparation of the manuscript.

**Competing interests:** Michael Koppitz and Michael Tripolt share a company founded after finishing the study to produce a consumer-usable early skin cancer detection app. This does not alter our adherence to PLOS ONE policies on sharing data and materials. The remaining authors have no conflicts of interest to declare.

lesion being malignant. In conjunction with the wide spread use of smartphones this new AI approach opens the opportunity for a higher early detection rate of skin cancer with consecutive lower epidemiological burden of metastatic cancer and reducing health care costs. This neural network moreover facilitates the empowerment of patients, especially in regions with a low density of medical doctors.

## Registration

Approved and registered at the ethics committee of the Medical University of Graz, Austria (Approval number: 30–199 ex 17/18).

## Introduction

The incidence of skin cancer, malignant melanoma (MM) and non-melanoma skin (NMSC), is rising worldwide. In Europe, over 144.000 new MM cases are reported each year, being responsible for more than 27.000 deaths per year [1, 2]. The most common NMSC are basal cell carcinomas (BCC) and squamous cell carcinomas (SCC). However, the exact count of NMSC in Europe is not definable as not all tumors are gathered in local databases. Data from Germany suggest an incidence of 119-145/100.00 in 2010 [3, 4]. Both, BCCs and SCCs usually do have a favorable prognosis, but also have the potential for local destructive growth and in advanced cases also for metastatic disease. The reported metastases rate of BCC ranges from 0.0029% to 0.55% with common sites being the regional lymph nodes, lunges, bones, skin, and the liver. Focusing on SCC, it is reported that approximately 4% of all patients will develop metastases and 1.5% die from the disease [5–9]. Recent data from the American Academy of Dermatology [10] estimates that NMSC affects more than 3 million Americans per year and that 196.060 new cases of melanoma were diagnosed in 2020.

Although there have been advancements in the treatment of metastatic skin cancer in the last decade, the mortality rates, especially those of MM, still strongly depend on its early detection [11–13]. While the 5-year survival rate according to the AJCC-Classification 8 (American Joint Committee on Cancer) is nearly 100% for very thin melanomas, it is less than 30% for advanced stages. Consequently, early detection of skin cancer is crucial to avoid metastatic disease as well as high morbidity and mortality rates. Of note, health care costs are another considerable factor that can be influenced by early detection. A recent Australian study showed yearly average costs of 115.109 AUS$ per case for metastatic melanoma; in contrast, the yearly costs for the early stages 0–1 are about 1681 AUS$ [14] on average.

There is growing evidence that artificial intelligence is a valuable supplementary tool in various medical sectors (e.g., radiology and dermatology) [15, 16]. The emergence of new technological tools, especially convolutional neural networks (CNNs), enabled an automated, in vitro image-based diagnosis of various skin diseases [17]. Several studies [18–28] investigated CNNs regarding their diagnostic accuracy concerning melanoma recognition. Notably, most skin cancer recognition networks have currently been used for the classification of high-quality images. However, in a realistic scenario a high variance of image quality and image characteristics have to be taken into account. Very recently, a meta-analysis [29] reported an unreliable performance for smartphone-based applications; the application with the best performance had a sensitivity of 80% and specificity of 78%.

## Neural networks

We used a classical convolutional neural network (CNN) and a novel stratification CNN based on a region proposal network (RPN). Both are usable on smartphones in daily routine. Notably, both algorithms were previously developed, already fulfill the CE-criteria and are registered as medical product at the Austrian Federal Office for Safety in Health Care (registration number: 10965455). The effectiveness of similar neural networks was recently demonstrated in an "in-vivo" skin cancer classification task [28]. The RPN is more hardware demanding and needs, in today available smartphones, a longer processing time (2–5 seconds); but this is still acceptable to analyze the lesions. More detailed information on the algorithms is given in the supplementary material.

The aim of the study presented herein was to evaluate and validate the diagnostic accuracy in skin cancer recognition of the two different neural networks (image classifier/analyze and region proposal network / detect), which were tested separately and in conjunction on a mobile phone in a clinical setting. This constitutes a novel approach in optical clinical detection of skin lesions.

## Methods

### Study design

The study was prospective and designed as single-center study at the Department of Dermatology and Venereology, a tertiary referral center, in Graz in order to evaluate the diagnostic and risk-assessment accuracy of two CNNs in comparison to the histopathological and clinical diagnosis. The study was approved by the local ethics committee (Approval number: 30–199 ex 17/18). All procedures were conducted according to the principles of the Declaration of Helsinki and patients gave written informed consent prior to enrollment. The risk classification of the CNN was stated as correct, if it matched the clinical diagnosis of two dermatologists or the histological diagnosis if available. Notably, the histological diagnosis (in case a biopsy/excision was done) was always considered as reference standard. However, if both dermatologists evaluated a lesion as being benign (based on distinct clinical and dermoscopic malignancy-criteria), no histology was performed—this procedure explains the low number of histologically proven lesions compared with the number of included lesions. Consequently, the last decision (whether to biopsy/excise a lesion or not) was always made by the two dermatologists, even if the algorithm made an opposite risk classification. A small proportion of patients denied biopsy/excision of a suspicious lesions. In these cases, the diagnosis of the two dermatologists was considered as reference.

Moreover, a clinical and dermoscopic digital follow-up of all non-excised "dysplastic nevi" was performed 3–6 months after evaluation.

### Participants

Patients with a minimum age of 18 years, who were either scheduled for preventive skin examination or removal of at least one skin lesion, were eligible for participation. The recruitment process was consecutive, meaning that every patient, who was seen by one of the study authors was ask to participate. The only exclusion criterion was a residual tumor after incomplete resection of any skin cancer. These broadly defined inclusion criteria are based on the fact that the study was conducted at a tertiary referral center usually caring for patients at high risk for developing skin cancer of any type. The enrollment phase was between June 2018 and December 2019 and five lesions (both, benign and malignant appearing) were scanned per patient averagely.

## Procedure and mobile devices

In a first step, patients were screened by at least two experienced dermatologists independently of each other clinically and dermoscopically. The lesions´ evaluation was made consensually. Scans of nails and scalp-hair were excluded. Notably, also clearly benign lesions were selected for further AI-evaluation. Second, images of the selected lesions were taken by a third dermatologist using the integrated camera and the flash of different mobile phones (Samsung S7, Honor 7A, iPhone Xs, iPhone 6s, Huawei P10) from a distance of approximately 15cm. For an adequate blinding-process, the evaluating dermatologists in the first step had no knowledge of the algorithms´ evaluation. As the detect algorithm was in development until March 2019, the first 132 patients were assessed only with the analyze algorithm.

## Algorithms

We tested two different machine-learning algorithms, named analyze and detect. During their developing phase, they were trained with 18.384 images labeled with one of 47 distinct subcategories. Both algorithms are based on a two-step approach: 1) calculating probabilities for the different labels per image, and 2) risk assessment based on these probabilities.

Inspired by previous publications, we developed a three-level decision-tree (subcategory, category, risk level) (Fig 1). The 47 subcategories were divided into five categories (benign, anatomical structure, non-neoplastic, precancerous, malignant). These categories were then divided into three risk levels (low, medium, high). Detailed information about the algorithms is given in the supplementary.

## Statistical analysis

To quantify the risk-assessment accuracy of both algorithms, sensitivity and specificity of these systems in comparison to the histopathological (if available) and clinical diagnosis were calculated. In contrast to binary (healthy versus sick, high versus low risk) diagnostic tools, the studied approaches provided three risk levels in concordance with the clinical practice of assessing

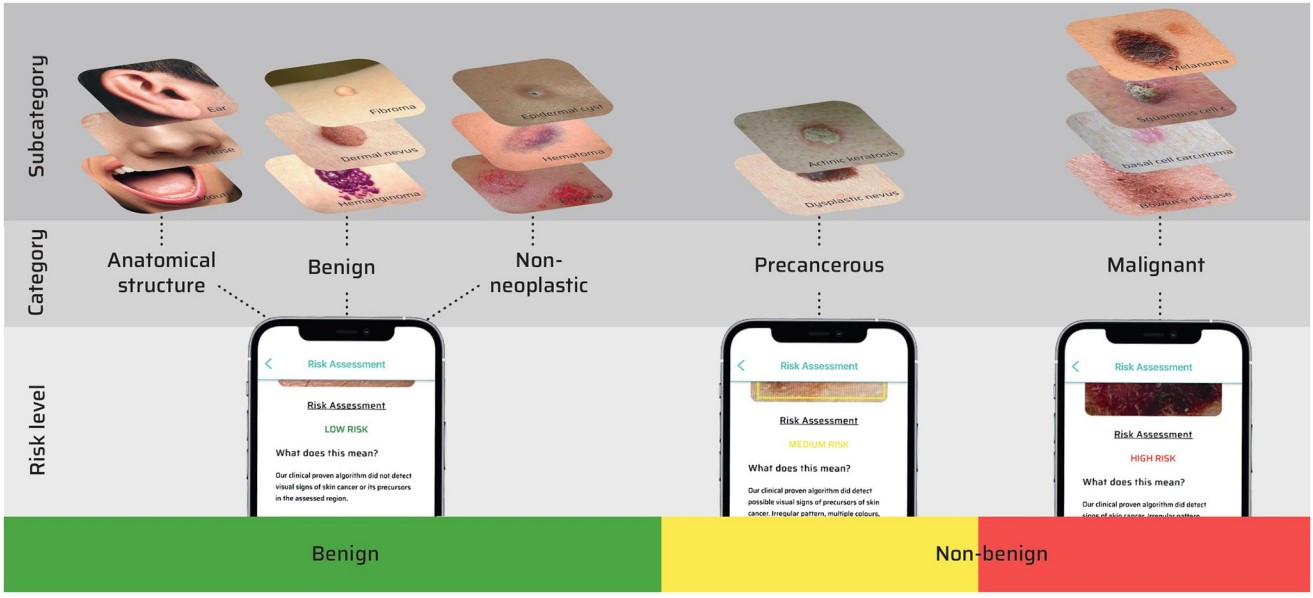

**Fig 1. Graphical depiction of the "three-level decision-tree".**

a skin lesion as benign, precancerous or malignant. To calculate the sensitivity and specificity the risk levels "medium" and "high" were summarized as "non-benign" and the risk level "low" was accordingly entitled "benign". This classification is line with the clinical relevance as both risk levels in the group "non-benign" need further medical action independently of their risk level. Statistical parameters were based on the risk level high and medium versus low to the endpoint benign vs. non-benign.

The specificity was calculated twice for each algorithm, one including and one excluding images of the risk category "benign". For binary analyses the lesions rated as "benign" were excluded in one calculation in order to adequately differentiate the risk categories "medium/yellow" and "high/red". Performance differences between both algorithms were assessed via 2-sample tests for equality of proportions with continuity correction based on Pearson's Chi-square statistic.

## Results

### Patients´ and lesions´ characteristics

A total of 238 patients with 1171 lesions (86 female; 36.13%) were included; all of them were scanned with the analyze algorithm and 92 patients (38.65%) with 552 lesions (27 female; 29.35%) were additionally screened with the detect algorithm. The average age was 66.19 (SD = 17.05) in the analyze test group and 66.42 (SD = 17) in the detect group. The distribution of the skin types according to Fitzpatrick scattered as follows: **Analyze**: Skin type I: 29 (12.2%); skin type II: 141 (59.2%); skin type III: 64 (26.9%); skin type IV: 4 (1.7%). **Detect**: Skin type I: 13 (14.1%); skin type II: 54 (58.7%); skin type III 22 (24%); skin type IV: 3 (3.2%). On average, 5 lesions per patient (range: 1 to 18) were selected and scanned. The detailed distribution is shown in Table 1.

Lesions allocated to the risk group "high" (n = 196): Malignant melanoma (n = 20), squamous cell carcinoma (n = 55), basal cell carcinoma (n = 114) and Bowen´s disease (n = 7).

Lesions allocated to the risk group "medium" (n = 283): Actinic keratosis (n = 115) and dysplastic nevus (n = 168).

The assignment of the lesions to the respective risk groups by the algorithm is given is given in the Tables 2–4. In 165 lesions (154 analyze and 67 detect) a histological examination was performed.

### Mobile devices

The mainly used operating system on the mobile devices was Android for both algorithms (analyze in 85.09% and detect in 85.74% of the cases). The remaining scans were performed with iOS. The imaging procedure was done easily in most of the cases excluding the abovementioned excluded body regions and usually no more than one attempt per lesion was

**Table 1. Distribution of the scanned lesions in the different age classes.** The total number of participants as well as the percentage in the respective algorithm-group is shown.

| Age in years | 20–29 | 30–39 | 40–49 | 50–59 | 60–69 | 70–79 | 80–89 | 90+ | Total |
|---|---|---|---|---|---|---|---|---|---|
| Analyze total (n) | 5 | 14 | 29 | 41 | 42 | 47 | 48 | 10 | 236* |
| Analyze relative (%) | 2.1 | 5.9 | 12.3 | 17.4 | 17.8 | 19.9 | 20.3 | 4.3 | 100.0 |
| Detect total (n) | 1 | 5 | 13 | 17 | 11 | 21 | 18 | 4 | 90* |
| Detect relative (%) | 1.1 | 5.6 | 14.4 | 18.9 | 12.2 | 23.3 | 20.0 | 4.5 | 100.0 |

*In two patients no information concerning their age was available explaining the different number of patients.

**Table 2. Assignment of the lesions to the respective risk groups by the algorithm.**

|  | Low | Medium | High | Histology |
|---|---|---|---|---|
| **Analyze n = 1155** | 638 | 287 | 230 | 154 |
| **Detect n = 552** | 321 | 145 | 86 | 67 |

* In 16 lesions the analysis did not work due to a software error.

needed in order to get a good image. The average imaging time including the recording of patient´s data was three minutes.

## Accuracy of the algorithms

**Sub-category accuracy.** When focusing on the 47 subcategories, the diagnostic accuracy of the detect algorithm was 88.35% (best subcategories [each exceeding 88.35%]: Bowen´s disease, comedo, hematoma, keloid; worst subcategory [50%]: dermatofibroma), compared to 81.74% in the analyze group (best subcategory [each exceeding 81.74%]: actinic keratosis, hemangioma, seborrheic keratosis, cyst; worst subcategory [33.3%]: hypopigmentation) (p = 0.0005, $\chi^2$ = 12.05). Fig 2 shows detailed information on the diagnostic accuracy of both algorithms concerning the 47 subcategories summarized in their respective risk levels.

Concerning the operating systems, analyze showed an average diagnostic accuracy of 81.2% on Android and 84.83% on iOS (p = 0.29, $\chi^2$ = 1.11). For detect, the average diagnostic accuracy was 88.24% for Android and 89.02% for iOS (p = 0.98, $\chi^2$ = 0).

**Overall accuracy.** Statistical analysis in the classification groups "benign" versus "non-benign" showed a sensitivity of 95.35% (CI 93.45–97.25) for analyze and 96.4% (CI 93.94–98.85) for detect. The specificity, including non-neoplastic lesions, increased to 90.32% (CI 88.1–92.54) for analyze and 94.85 (CI 92.46–97.23) for detect. No significant difference between the sensitivity (p = 0.6) could be proven between both algorithms. However, we found a significant difference between the algorithms when focusing on their specificity (including non-neoplastic lesions p = 0.02, excluding non-neoplastic lesions p = 0.04).

A notable lower diagnostic accuracy (76% with detect and 72% with analyze) for malignant lesions compared to benign lesions was also observed and mostly (over 70%) attributable to incorrect diagnoses within the malignant category.

## Discussion

Our results indicate that both tested networks have the potential to serve as cost and time-effective skin screening tools for the general population. Due to the substantial gain of processing power and camera quality, most mobile devices fulfill the technological requirements to enable mobile skin screenings at home (Fig 3).

The use of artificial intelligence in this context has certain advantages like cost- and time-savings [30]. Of note, neural networks cannot replace a dermatologist, mostly because these systems are not able to take further diagnostic and/or therapeutic measures [30]. However, these systems are a promising concept to alert patients with (early) skin cancer. Consequently, the patients can consult a dermatologist earlier in order to avert further harm.

Both networks were trained with identical datasets and archived similarly good results, despite differences in the network structure. Test- and training-images were taken with a variety of mobile devices, in different settings and by non-trained individuals making the algorithms robust against low image quality and variations.

**Table 3.** Shown are the absolute numbers of the six subcategories "red/high" and "yellow/medium" with their allocation to the three risk groups by each algorithm (a and b). 3c shows the histopathological results of all excised lesions. 3d and 3e display crosstabulations of the respective algorithm with the clinical category.

a) Absolute numbers of the algorithm "ANALZYE" (% in brackets) for the risk groups "red" and "yellow"

|  |  | Risk Groups | | | | Total |
|---|---|---|---|---|---|---|
|  |  | Missing values* | High | Low | Medium |  |
| Diagnostic_category | BCC** | 3 (2.6%) | 105 (92.1%) | 6 (5.3%) | 0 (0.0%) | 114 |
|  | Bowen´s disease | 1 (14.3%) | 5 (71.4%) | 1 (14.3%) | 0 (0.0%) | 7 |
|  | Melanoma | 1 (5.0%) | 16 (80.0%) | 1 (5.0%) | 2 (10.0%) | 20 |
|  | SCC*** | 1 (1.8%) | 44 (80.0%) | 1 (1.8%) | 9 (16.4%) | 55 |
|  | Actinic keratosis | 0 (0.0%) | 23 (20.0%) | 4 (3.5%) | 88 (76.5%) | 115 |
|  | Dysplastic nevus | 0 (0.0%) | 19 (11.3%) | 9 (5.4%) | 140 (83.3%) | 168 |
| Total |  | 6 (1.3%) | 212 (44.2%) | 22 (4.6%) | 239 (49.9%) | 479 |

b) Absolute numbers of the algorithm "DETECT" (% in brackets) for the risk groups "red" and "yellow"

|  |  | Risk Groups | | | | Total |
|---|---|---|---|---|---|---|
|  |  | Missing values* | High | Low | Medium |  |
| Diagnostic_category | BCC** | 66 (57.9%) | 43 (37.7%) | 4 (3.5%) | 1 (0.9%) | 114 |
|  | Bowen´s disease | 5 (71.4%) | 2 (28.6%) | 0 (0.0%) | 0 (0.0%) | 7 |
|  | Melanoma | 7 (35.0%) | 13 (65.0%) | 0 (0.0%) | 0 (0.0%) | 20 |
|  | SCC*** | 31 (56.4%) | 20 (36.4%) | 0 (0.0%) | 4 (7.2%) | 55 |
|  | Actinic keratosis | 72 (62.6%) | 2 (1.7%) | 1 (0.9%) | 40 (34.8%) | 115 |
|  | Dysplastic nevus | 76 (45.2%) | 1 (0.6%) | 3 (1.8%) | 88 (52.4%) | 168 |
| Total |  | 257 (53.6%) | 81 (16.9%) | 8 (1.7%) | 133 (27.8%) | 479 |

c) Absolute numbers and results of the histolopathological reports (% in brackets) for the risk groups "red" and "yellow"

|  |  | Histology | | | | | Total |
|---|---|---|---|---|---|---|---|
|  |  | SCC | MM | BCC | Actinic keratosis | Scar |  |
| Clinical_diagnosis | BCC | 3 (2.9%) | 1 (1.0%) | 97 (95.1%) | 0 (0.0%) | 1 (1.0%) | 102 |
|  | MM**** | 0 (0.0%) | 13 (100.0%) | 0 (0.0%) | 0 (0.0%) | 0 (0.0%) | 13 |
|  | SCC | 38 (76.0%) | 0 (0.0%) | 10 (20.0%) | 2 (4.0%) | 0 (0.0%) | 50 |
| Total |  | 41 (24.8%) | 14 (8.5%) | 107 (64.8%) | 2 (1.2%) | 1 (0.7%) | 165 |

d) Crosstabulation of the algorithm "ANALZYE": ANALYZE_risk* clinical category

| ANALYZE risk |  | Clinical Category | | | Total |
|---|---|---|---|---|---|
|  |  | Benign | Malignant | Precancerous |  |
|  | Missing* | 10 | 6 | 0 | 16 |
|  | High | 18 | 170 | 42 | 230 |
|  | Low | 616 | 9 | 13 | 638 |
|  | Medium | 48 | 11 | 228 | 287 |
| Total |  | 692 | 196 | 283 | 1171 |

e) Crosstabulation of the algorithm "DETECT": DETECT_risk* clinical category

| DETECT risk |  | Clinical Category | | | Total |
|---|---|---|---|---|---|
|  |  | Benign | Malignant | Precancerous |  |
|  | Missing* | 362 | 109 | 148 | 619 |
|  | High | 5 | 78 | 3 | 86 |
|  | Low | 313 | 4 | 4 | 321 |
|  | Medium | 12 | 5 | 128 | 145 |
| Total |  | 692 | 196 | 283 | 1171 |

* Missing values: In some cases, the algorithms did not assess the risk of a respective lesion due to a software error. The numbers are much higher in Table 3b as the algorithm "detect" was developed later (details are described in the running text)

** Basal cell carcinoma

*** Squamous cell carcinoma

**** Malignant melanoma

* Missing values: In some cases, the algorithm did not assess the risk for a respective lesion due to a software error.

* Missing: In some cases, the algorithm did not assess the risk for a respective lesion due to a software error. The numbers are much higher than in Table 3d as the DETECT algorithm was developed and tested later.

**Table 4. Crosstabulations risk*histology for ANALYZE (a) and DETECT (b).**

**a) Crosstabulations risk*histology for the algorithm "ANALYZE"**

| Risk | | Histology | | | | | Total |
|------|------|------|------|------|------|------|------|
| | | SCC** | MM*** | BCC**** | Actinic Keratosis | Scar | |
| | Missing* | 1 | 1 | 3 | 0 | 0 | 5 |
| | High | 37 | 13 | 98 | 2 | 1 | 151 |
| | Low | 1 | 0 | 6 | 0 | 0 | 7 |
| | Medium | 2 | 0 | 0 | 0 | 0 | 2 |
| **Total** | | 41 | 14 | 107 | 2 | 1 | 165 |

**b) Crosstabulations risk*histology for the algorithm "DETECT"**

| Risk | | Histology | | | | | Total |
|------|------|------|------|------|------|------|------|
| | | SCC** | MM*** | BCC**** | Actinic Keratosis | Scar | |
| | Missing* | 22 | 5 | 59 | 1 | 1 | 88 |
| | High | 18 | 9 | 43 | 1 | 0 | 71 |
| | Low | 0 | 0 | 4 | 0 | 0 | 4 |
| | Medium | 1 | 0 | 1 | 0 | 0 | 2 |
| **Total** | | 41 | 14 | 107 | 2 | 1 | 165 |

* Missing risk value: In some cases, the algorithm did not assess the risk for the respective lesion due to a software error.

* Missing risk value: As the „ANALYZE" algorithm evaluation started before the "DETECT" algorithm was introduced, some lesions were only assessed by "ANALYZE" explaining missing risk values for "DETECT".

** Squamous cell carcinoma

*** Malignant melanoma

**** Basal cell carcinoma

Several conceptual and practical differences between both networks have to be noted. A broad range of evidence supports whole-image analysis [15] and support vector machines [31]. The new approach of the use of RPNs in the context of skin cancer diagnosis has certain advantages. We showed that the RPN-based detect algorithm diagnoses were significantly more accurate than the whole-image analyze algorithm approach. A reason for the higher diagnostic accuracy might be a better performance for images with several or disjoint lesions. As every lesion is evaluated separately, smaller (pre-) cancerous lesions can be detected even nearby larger benign lesions. Nevertheless, this benefit was not conferrable to the benign vs. non- benign assessment. Another advantage of RPNs is, that they draw bounding boxes, highlighting the analyzed sites of the image, and hereby allowing users to confirm that the favored lesion was analyzed (Fig 4). The only drawback is, that detect requires more CPU capacities. In up-to-date smartphones it takes several seconds to analyze a picture. In contrast analyze is less hardware demanding and capable of real-time classification even as a video stream.

Some limitations of this study have to be mentioned: One major challenge for studies in a hospital setting [29–44] is posed by the diagnosis validation of clinically non-malignant lesions. Clinically malignant and most precancerous skin lesions are resected and histologically examined, whereas the diagnosis of clinically benign lesions is only based on visual evaluation. Given a visual classification-accuracy of below 90% (sensitivity 87% and specificity 81%) [37] in dermatologists, test- and training-data can be estimated to contain a number of misclassifications. We aimed to reduce this bias by validating clinically benign lesions by at least two dermatologists and by the follow-up of dysplastic nevi.

A second limitation is the generalizability of the study population in terms of their risk profile, age and skin type. The average age of our study population with over 65 years is

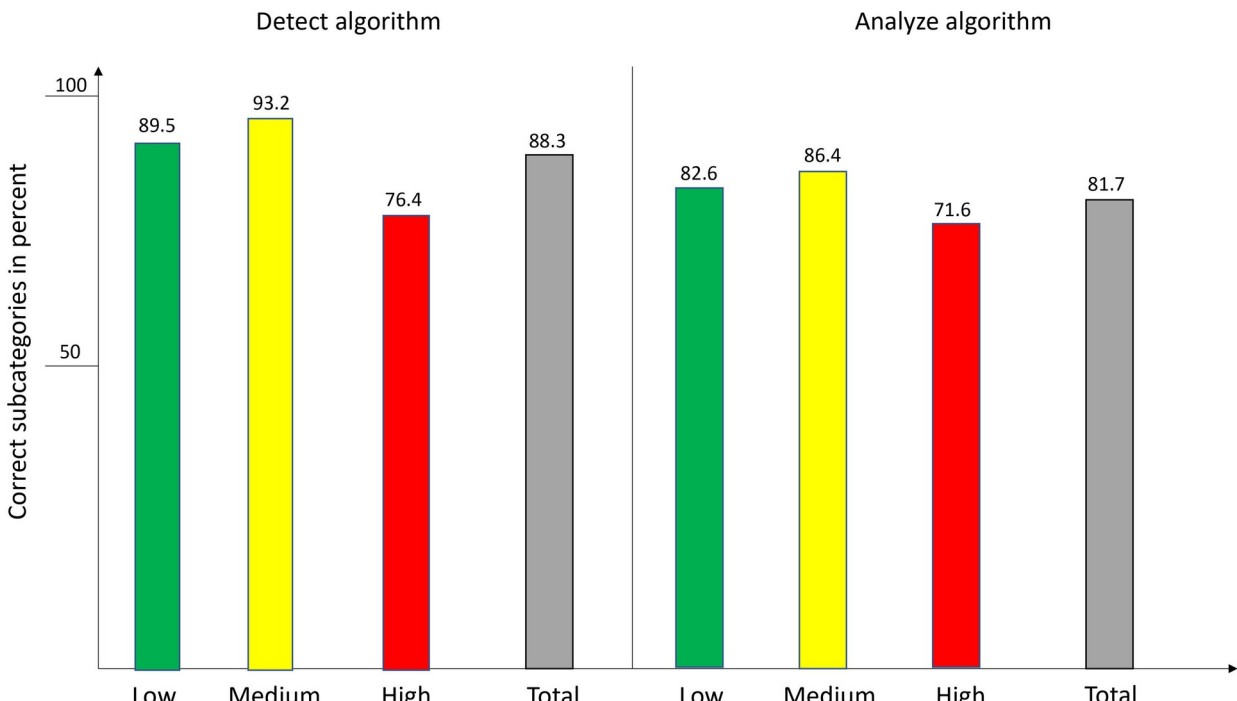

**Fig 2. Diagnostic accuracy of both algorithms in the three risk levels as well as the overall diagnostic accuracy.** The columns indicate the percentages of correct diagnoses in the respective risk level.

significantly higher than the average age of the Austrian population. In addition, both networks were trained and tested solely with lesions of the Central European population (skin type according to Fitzpatrick I-IV). The distribution of skin-types underrepresents individuals with dark skin (Fitzpatrick V+VI) [44] and requires an expansion of future training- and test-images. Additionally, we have chosen a highly selected population (mostly patients with a high risk for developing any kind of skin cancer), which is not fully applicable for the general population; however, our primary aim was to detect malignant lesions with a high sensitivity explaining this selected population. Our study investigated the effectiveness of two smartphone-compatible neural networks in the risk assessment of skin lesions. Both approaches were proven to be reliable and effective tools in skin cancer detection. A notable lower diagnostic accuracy (76% with detect and 72% with analyze) for malignant lesions compared to the other categories was mostly (over 70%) attributable to incorrect diagnoses within the malignant category. This confusion did therefore not influence the accuracy of the risk level assessment. The remaining lesions were wrongly labeled to the precancerous category, which also had no effect on the risk level assessment. These incorrect diagnoses/labels within the non-benign category (precancerous and malignant lesions) have no clinical relevance, as the allocation to this category needs further medical attention in any case. Furthermore, the only relevance for the patients is the differentiation between benign and non-benign lesions. In this context, the study by Tran et al. [45], who investigated the diagnostic accuracy of general practitioners and dermatologists in various skin conditions (e.g., benign and malignant skin tumors, bacterial/fungal infections, inflammatory diseases), should be mentioned. The overall diagnostic accuracy of the general practitioners in their study was much lower than those of neural networks in our study. Obviously, neural networks will never achieve a diagnostic

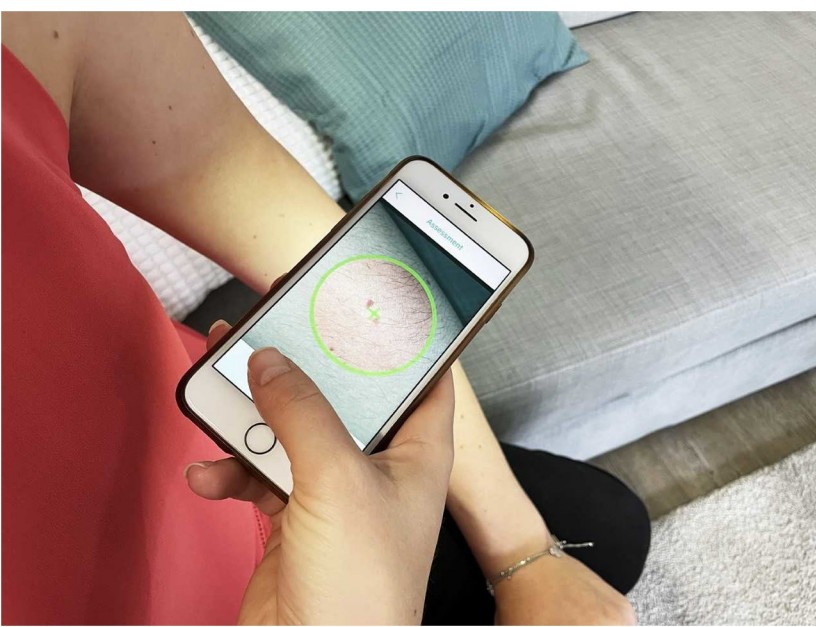

**Fig 3. Easy use of the algorithm by the patient on the smartphone.**

accuracy of 100%; however, in comparison to abovementioned study, our neural networks surpassed the general practitioners; given that, our networks could have a positive impact on the health system as unnecessary visits and histological examination will be reduced [45, 46].

Despite above-mentioned limitations, these networks are a promising and market ready technological innovation with the potential to further increase awareness of skin cancer and promote its early detection. Thereby, the health and financial burden of skin cancer could be decreased for the patients and the society.

## Conclusion

Our study showed both tested AI-approaches (CNN and RPN) to be capable of classifying images of various skin lesions with a high accuracy regarding the subcategory diagnosis and the risk assessment. Three main parameters of interest where examined: The diagnostic accuracy (showing the efficacy of the AI), the benign vs. non- benign sensitivity and specificity (showing the clinical relevance). Concerning the overall diagnostic accuracy, the detect-algorithm outperformed the analyze-algorithm significantly (88% versus 82%). In terms of risk assessment, no significant differences were found between the two approaches concerning the sensitivity (each exceeding 95%); however, when focusing on the respective specificity the detect algorithm outperformed the analyze algorithm (94.85% versus 90.43%). These results could alert patients with malignant lesions to consult a physician quickly. Whereas, especially in times of the Covid-19 pandemic, patients with benign lesions are prevented from doing so. In conclusion, both algorithms surpassed the performance of automated classifications [14, 32–34] and assessments by physicians [32–34, 47, 48] in comparable classification tasks. This neural network moreover facilitates the empowerment of patients, especially in regions with a low density of medical doctors.

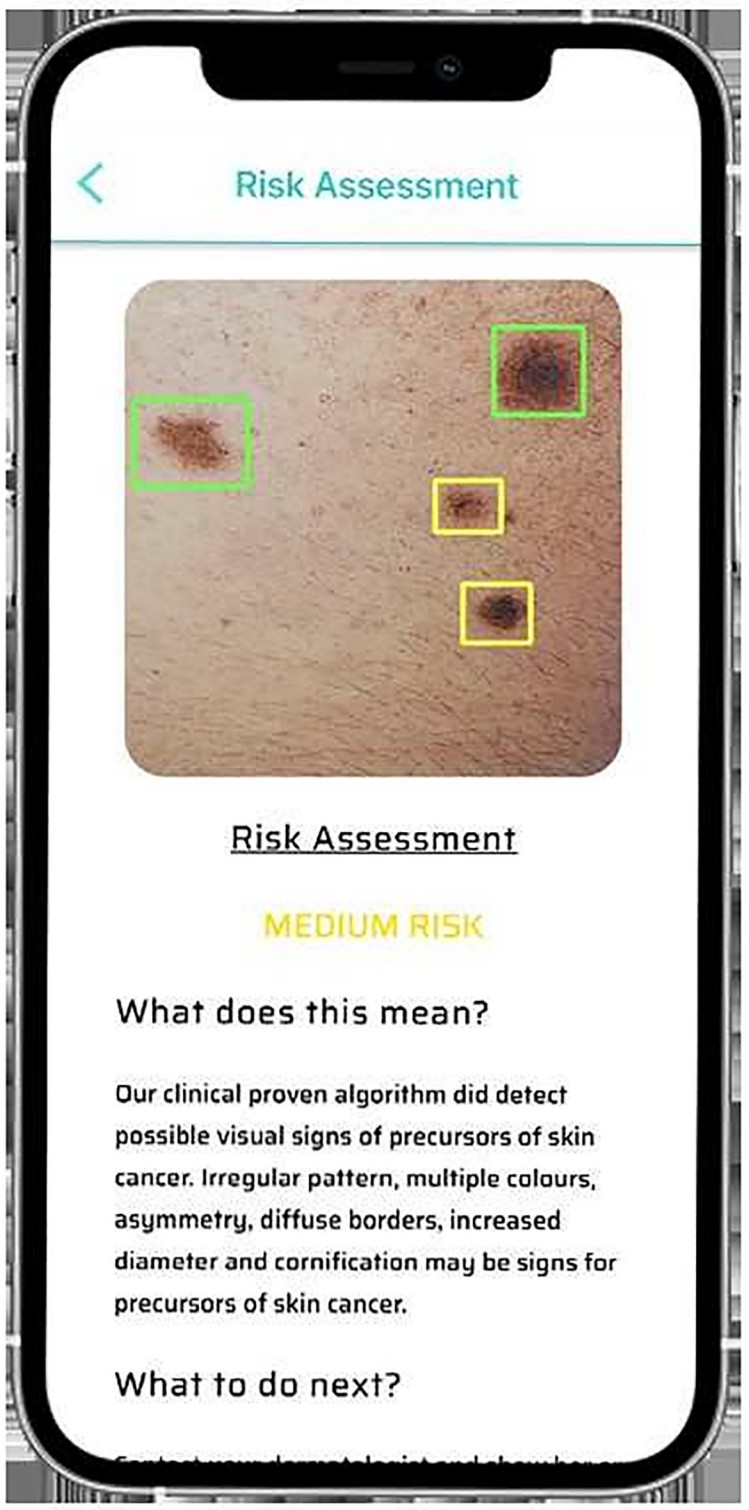

**Fig 4. The RPN-based detect algorithm with bounding boxes highlighting the four analyzed lesions of one image.** Every detect lesion is assigned to a risk level as indicated by different colors.

## Supporting information

**S1 File.**
(DOCX)

**S1 Data.**
(XLS)

## Author Contributions

**Conceptualization:** Michael Tripolt.

**Data curation:** Teresa Kränke, Katharina Tripolt-Droschl, Lukas Röd, Rainer Hofmann-Wellenhof, Michael Koppitz, Michael Tripolt.

**Formal analysis:** Lukas Röd, Rainer Hofmann-Wellenhof, Michael Koppitz.

**Investigation:** Katharina Tripolt-Droschl, Michael Tripolt.

**Project administration:** Rainer Hofmann-Wellenhof, Michael Tripolt.

**Software:** Michael Koppitz.

**Supervision:** Teresa Kränke.

**Validation:** Teresa Kränke.

**Writing – original draft:** Teresa Kränke.

**Writing – review & editing:** Katharina Tripolt-Droschl, Rainer Hofmann-Wellenhof, Michael Koppitz, Michael Tripolt.

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
