## [Decision Letter · Decision Letter 0]

12 Apr 2022

PONE-D-22-08100New AI-algorithms on smartphones to detect skin cancer in a clinical settingPLOS ONE

Dear Dr. Kränke,

Thank you for submitting your manuscript to PLOS ONE. After careful consideration, we feel that it has merit but does not fully meet PLOS ONE’s publication criteria as it currently stands. In accordance with the expert reviewers, I have a number of both technical and conceptual concerns. Rather than repeat those points here, I refer you to the specific remarks (below) for details. Therefore, we invite you to submit a revised version of the manuscript that addresses the points raised during the review process.

We look forward to receiving your revised manuscript.

Kind regards,

Nikolas K. Haass, MD/PhD

Academic Editor

PLOS ONE

Journal Requirements:

"Michael Koppitz and Michael Tripolt share a company founded after finishing the study to produce a consumer-usable early skin cancer detection app. The remaining authors have no conflicts of interest to declare. "

Additional Editor Comments:

In accordance with the expert reviewers, I have a number of both technical and conceptual concerns. Rather than repeat those points here, I refer you to the specific remarks (below) for details. 

Reviewers' comments:

Reviewer's Responses to Questions

**Comments to the Author**

1. Is the manuscript technically sound, and do the data support the conclusions?

Reviewer #1: Partly

Reviewer #2: Partly

2. Has the statistical analysis been performed appropriately and rigorously? 

Reviewer #1: I Don't Know

Reviewer #2: I Don't Know

3. Have the authors made all data underlying the findings in their manuscript fully available?

Reviewer #1: No

Reviewer #2: No

4. Is the manuscript presented in an intelligible fashion and written in standard English?

Reviewer #1: Yes

Reviewer #2: Yes

5. Review Comments to the Author

Reviewer #1: This paper describes an evaluation of two AI algorithms for skin cancer diagnosis . Unfortunately it is not well reported and does not meet available reporting guideline for test accuracy studies. I recommend that the study (and abstract) are at least reported in accordance with the STARD reporting guideline. Unfortunately the extension to cover AI tools is not yet published, however the authors could also consider looking at the MI-CLAIM checklist for guidance on what is needed for reporting of an AI tool (https://doi.org/10.1038/s41591-020-1041-y). It is actually quite difficult to tell whether the authors consider the paper tor report both development and validation of the AI tools or validation only – there is a paucity of detail in regard to several aspects of the study making it really quite difficult to review. It seems possible that plans to commercialise the AI algorithms may have led to details being withheld from the report. Given the potential impact of using these type of algorithm on patients I would encourage authors to report sufficient details of both the development and validation of the tools to allow the methods and results to be fully judged and to support any subsequent utilisation of these tools.

Title – should reflect that this is diagnostic test accuracy study, please reword

Abstract – please refer to the STARD extension for Abstracts. Taking this as a report of a validation study only, then missing items include details about eligibility criteria for participants and lesions, setting (ie it is a dermatology referral centre), participant recruitment method, and description of the index test and reference standard. The Results should report the number of ‘cases’ amongst the evaluated lesions, and neither Results nor Conclusions adequately report that the AI tools are intended to assess the risk of a skin lesion being malignant.

Introduction

Line 2 – I believe that ‘keratinocyte’ skin cancer is now the more generally accepted term for ‘non-melanoma’

Line 8 – Could add caveat that BCC and SCC have a ‘generally’ favourable prognosis or similar

Line 25 – The Rezazade Mehrizi reference (15) is not the most appropriate reference to support the statement about potential value of AI tools across sectors

Line 35-40 – This paragraph about programming of a CNN and novel stratification CNN implies that the development phase for the AI tools evaluated in this study has already been conducted, is that correct? Very limited detail is provided about the development, training and validation of the two models either here or in the rest of the paper or supplement. Reference 28 (Han 2018) cited in this paragraph does not have any authors in common with this paper so presumably reports a similar method rathe than the development of either model used here. It is essential that the development of AI tools is transparently reported so that readers can be satisfied that such tools have been rigorously and robustly evaluated (this equally applies to tools intended for commercial use).

Methods

A number of standard elements for any DTA evaluation are missing from this section. I have highlighted a few but suggest the authors refer to the STARD reporting guideline for further information about this.

Line 50-51 – State that histology or clinical diagnosis was the reference standard; more details needed in regard to selection for histology – was this independent of the result of the two index tests or not? If so, how was this implemented? Discussion also mentioned follow up of dysplastic nevi, why was is this not mentioned in the methods?

Line 57-58 – There is potentially quite a difference between participants scheduled for preventive skin examination (who may have no suspicious/biopsied skin lesions) and those who were scheduled for removal of at least one skin lesion. Please explain more about how participants were included/excluded and whether recruitment was consecutive or not. For the former group in particular, did all participants contribute at least one suspicious lesion? How many ‘benign appearing’ lesions were sampled per patient? It is very important to understand the potential for selection bias and the spectrum of included participants in order to judge the applicability of results.

Line 62-69 – This section describes both the reference standard and acquisition of images for the index tests – these should be separate processes that are described separately and in more detail.

Line 62-64 More information is needed about how the clinical diagnoses were made and the result recorded (did the same two dermatologists see all patients?) as this forms part of the reference standard. How were lesions selected for biopsy? How was the reference standard result categorised as ‘diseased’ versus ‘not diseased’, did diseased include clinical diagnosed BCCs for example, or were only histologically confirmed malignancies considered as disease positive.

Line 64-69. How were lesions selected for imaging? How did clinicians decide which of the several smartphones to use for imaging? Did the same two clinicians carry out the imaging as those who made the clinical diagnoses. Lack of blinding between index and reference standard and between the two index tests is potentially a major source of bias here.

Line 71-80 and Supplement – Detail of algorithm development is too meagre to really allow any comment to be made.

Line 81 Statistical analysis – What sample size considerations were made for the study?

Line 92-93 What is the justification for excluding lesions rated as ‘benign’ from specificity calculations?

Results

No information is provided about the participants other than age group – additional demographic and clinical characteristics are potentially very important for skin cancer diagnosis,

Importantly the final lesion diagnoses were not reported so the reader has no idea how many lesions had a final ‘malignant’ diagnosis nor what type of lesions the authors included in this definition. From the 95% CIs reported, over 300 malignant lesions must have been included in the sample which does imply a highly selected sample.

No information is given about how successful the imaging was (how many attempts needed to get a good image), nor any failure rate for the AI tools.

No absolute numbers to support the results are provided, only accuracy, sensitivity and specificity – this needs to be much more detailed to allow results to be checked and to provide more informative results.

Discussion – In my opinion, without the further detail requested in regard to the study methods and results it is not really possible to comment on the Discussion and author conclusions however it does appear that the significance of the results have been rather over-stated.

Reviewer #2: The article is interesting, innovative and relevant, but some important information to analyze the data is missing.

I would suggest you make a Table with the frequency of images in each of the five categories and how many of them had the clinical evaluation or histopathological report. Also it was not clear if you considered in your results all of the 238 patients or only those with biopsy or clinical evaluation?

Another question: did the dermatologists classified the images in low, medium and high risk or they were assigned automatically according to the diagnosis?

The agreement rate of diagnostic and risk classification could be stratified according to the lesions that were seen by dermatologists and had the histopathologycal report too. It is critical to know how the ground truth is being assigned to compare with the algorithm´s accuracy.

Reference 20 is equal to 22.

6. PLOS authors have the option to publish the peer review history of their article (what does this mean?). If published, this will include your full peer review and any attached files.

Reviewer #1: No

Reviewer #2: **Yes: **Mara Giavina Bianchi

---

## [Author Response · Author response to Decision Letter 0]

14 Jun 2022

RE: Submission of the revised version of the manuscript number PONE-D-22-08100

Dear Editor, 

Thank you for giving us the opportunity to re-submit our manuscript titled “New AI-algorithms on smartphones to detect skin cancer in a clinical setting”. Please find below the point-to-point answers to the suggestions made by the reviewers. 

Reviewer #1:

This paper describes an evaluation of two AI algorithms for skin cancer diagnosis. Unfortunately, it is not well reported and does not meet available reporting guideline for test accuracy studies. I recommend that the study (and abstract) is at least reported in accordance with the STARD reporting guideline. Unfortunately, the extension to cover AI tools is not yet published, however the authors could also consider looking at the MI-CLAIM checklist for guidance on what is needed for reporting of an AI tool (https://doi.org/10.1038/s41591-020-1041-y). It is actually quite difficult to tell whether the authors consider the paper tor report both development and validation of the AI tools or validation only – there is a paucity of detail in regard to several aspects of the study making it really quite difficult to review. It seems possible that plans to commercialise the AI algorithms may have led to details being withheld from the report. Given the potential impact of using this type of algorithm on patients I would encourage authors to report sufficient details of both the development and validation of the tools to allow the methods and results to be fully judged and to support any subsequent utilisation of these tools.

Authors response: We thank the reviewer for these comments. We have not done a developing/evaluation study, but rather a study on the clinical performance of two CE-certified algorithms using artificial intelligence. In their developing phase the algorithms were trained on 18.384 images labeled with one of 47 distinct subcategories in order to accurately distinguish benign and malignant skin lesions. Please refer to the supplement for further information and to the lines 35-49. 

We have referred to the STARD guidelines wherever we considered useful. 

Furthermore, we have provided details about function principles of both models in the supplementary material. As both algorithms are already CE-certified, the subject of our study was the validation / clinical performance of both modes; we therefore did not provide more information about the development or function principles. 

Title – should reflect that this is diagnostic test accuracy study, please reword

Abstract – please refer to the STARD extension for Abstracts. Taking this as a report of a validation study only, then missing items include details about eligibility criteria for participants and lesions, setting (ie it is a dermatology referral centre), participant recruitment method, and description of the index test and reference standard. The Results should report the number of ‘cases’ amongst the evaluated lesions, and neither Results nor Conclusions adequately report that the AI tools are intended to assess the risk of a skin lesion being malignant.

Authors response: We thank the reviewer for these comments. 

Title: We reworded it according to your suggestion. Abstract: Our study is a proof of concept of AI-algorithms already fulfilling the CE-criteria and is registered as medical product at the Austrian Federal Office for Safety in Health Care (registration number: 10965455). Consequently, we did not perform an evaluation study in the narrower sense, but rather a validation study/testing of the performance of already certified AI algorithms in a clinical setting. However, we referred to the STARD guidelines wherever we considered useful. Concerning your request on a more detailed description of participants´ recruitment and lesions´ eligibility criteria, we have added the missing information. In order not to exceed the abstract, we have made added the number of “cases” in the main manuscript and made a table. We reworded the conclusion according to your suggestions. 

Introduction

Line 2 – I believe that ‘keratinocyte’ skin cancer is now the more generally accepted term for ‘non-melanoma’ 

Line 8 – Could add caveat that BCC and SCC have a ‘generally’ favourable prognosis or similar

Line 25 – The Rezazade Mehrizi reference (15) is not the most appropriate reference to support the statement about potential value of AI tools across sectors

Line 35-40 – This paragraph about programming of a CNN and novel stratification CNN implies that the development phase for the AI tools evaluated in this study has already been conducted, is that correct? Very limited detail is provided about the development, training and validation of the two models either here or in the rest of the paper or supplement. Reference 28 (Han 2018) cited in this paragraph does not have any authors in common with this paper so presumably reports a similar method rather than the development of either model used here. It is essential that the development of AI tools is transparently reported so that readers can be satisfied that such tools have been rigorously and robustly evaluated (this equally applies to tools intended for commercial use). 

Authors response: We thank the reviewer for these comments. 

Line 2: In Europe the term „non-melanoma skin cancer“ is well established as it not only includes tumors of keratinocyte origin (e.g., squamous cell carcinomas), but also 

others like Merkel cell carcinomas or basal cell carcinomas. Following this, we didn´t change this phrase as suggested by you. 

Line 8: We have changed this sentence accordingly.

Line 25: We have added an additional reference to adequately support the statement about AI as helpful tool. See new reference number 16 (Tomas et al.) (see line 15/16).

Line 35-40: The tested AI-algorithms are in line with the CE-criteria and registered as medical product at the Austrian Federal Office for Safety in Health Care (registration number: 10965455). We performed the first evaluation study of these algorithms in a clinical setting with “high-risk” patients at one tertiary referral center; hence, the intention of our study was to evaluate the performance of these algorithms in a clinical setting. Considering this, we did not provide many data about the development and training of the two models. The method of reference 28 was modified for our study purposes and was not topic of this paper.

Methods

A number of standard elements for any DTA evaluation are missing from this section. I have highlighted a few but suggest the authors refer to the STARD reporting guideline for further information about this.

Line 50-51 – State that histology or clinical diagnosis was the reference standard; more details needed in regard to selection for histology – was this independent of the result of the two index tests or not? If so, how was this implemented? Discussion also mentioned follow up of dysplastic nevi, why was is this not mentioned in the methods? 

Line 57-58 – There is potentially quite a difference between participants scheduled for preventive skin examination (who may have no suspicious/biopsied skin lesions) and those who were scheduled for removal of at least one skin lesion. Please explain more about how participants were included/excluded and whether recruitment was consecutive or not. For the former group in particular, did all participants contribute at least one suspicious lesion? How many ‘benign appearing’ lesions were sampled per patient? It is very important to understand the potential for selection bias and the spectrum of included participants in order to judge the applicability of results.

Line 62-69 – This section describes both the reference standard and acquisition of images for the index tests – these should be separate processes that are described separately and in more detail. 

Line 62-64 More information is needed about how the clinical diagnoses were made and the result recorded (did the same two dermatologists see all patients?) as this forms part of the reference standard. How were lesions selected for biopsy? How was the reference standard result categorised as ‘diseased’ versus ‘not diseased’, did diseased include clinical diagnosed BCCs for example, or were only histologically confirmed malignancies considered as disease positive. 

Line 64-69. How were lesions selected for imaging? How did clinicians decide which of the several smartphones to use for imaging? Did the same two clinicians carry out the imaging as those who made the clinical diagnoses. Lack of blinding between index and reference standard and between the two index tests is potentially a major source of bias here. 

Line 71-80 and Supplement – Detail of algorithm development is too meagre to really allow any comment to be made.

Line 81 Statistical analysis – What sample size considerations were made for the study? 

Line 92-93 What is the justification for excluding lesions rated as ‘benign’ from specificity calculations? 

Authors response: We thank the reviewer for these comments. 

We referred to the STARD guidelines wherever we considered useful.

Line 50-51: Histology was the reference standard; we have added this point. If both dermatologists evaluated a lesion as being benign, no histology was performed; consequently, the last decision (whether to biopsy/excise a lesion or not) was always made by the two dermatologists, even if the algorithm made an opposite risk classification. The follow-up of non-excised dysplastic nevi was added to this paragraph (see lines 59-68).

Line 57-58: We partially agree with you in that point; however, as we are a tertiary referral center, we have already “pre-selected” patients; meaning that we usually see patients at high risk for developing skin cancer of any type (e.g., patients with multiple and/or atypical nevi or patients with severe sun-damaged skin). Considering this, our study population was not that heterogeneous. We have adapted the paragraph (see lines 73-75).

No, not every patient from the group “scheduled for preventive skin examination” had a suspicious lesion to be biopsied/excised. Five lesions (both, benign and malignant) were scanned per patient averagely – we have already mentioned that in the results section and additionally added this information to this paragraph (see line 76).

Line 62-69: We have provided this information in the supplement. 

Line 62-64: The two dermatologists did a state-of-the-art clinical/dermatological examination screening the entire skin, the palms and soles and the scalp. However, this clinical examination was not always done by the same two dermatologists. As the clinical examination was performed by experienced dermatologists, they selected the lesions for biopsy based on distinct clinical/dermoscopic malignancy-criteria. As we have mentioned above, the last decision was always made by the clinicians regardless of the algorithm’s risk classification. The histology was always considered as reference standard, however, a small proportion of patients denied biopsy/excision of a suspicious lesions. In these cases, the diagnosis of the two dermatologists was considered as reference; we have added this point (see lines 61-66). 

Line 64-69: The lesions for imaging were randomly selected by the evaluating dermatologists. Notably, no one, who did the index testing has screened the patients. 

Line 71-80: As already mentioned, we performed an evaluation study of an already registered medical product fulfilling CE-criteria; further information concerning model training, evaluation and testing is given in the completed supplementary. 

Line 81: The sample size planning was performed by the Institute for medical statistics at the Medical University of Graz; we have added this information to the supplementary.

Line 92-93: We excluded the lesions rated as “benign” in one calculation in order to differentiate the groups “medium / yellow” and “high / red” adequately. We added this point (see lines 109-111). 

Results

No information is provided about the participants other than age group – additional demographic and clinical characteristics are potentially very important for skin cancer diagnosis. 

Importantly the final lesion diagnoses were not reported so the reader has no idea how many lesions had a final ‘malignant’ diagnosis nor what type of lesions the authors included in this definition. From the 95% CIs reported, over 300 malignant lesions must have been included in the sample which does imply a highly selected sample. 

No information is given about how successful the imaging was (how many attempts needed to get a good image), nor any failure rate for the AI tools. According to the used mobiles with a very good camera 

No absolute numbers to support the results are provided, only accuracy, sensitivity and specificity – this needs to be much more detailed to allow results to be checked and to provide more informative results. 

Discussion – In my opinion, without the further detail requested in regard to the study methods and results it is not really possible to comment on the Discussion and author conclusions however it does appear that the significance of the results have been rather over-stated.

Authors response: 

We thank the reviewer for these comments. Besides age groups, we have already provided data about the gender distribution. We additionally added the distribution of skin types throughout the study population (see lines 120-123). More information was not gathered per patient as this was not the primary subject of our study; moreover, the algorithms only evaluated the lesions without any patients´ information. 

We have provided the detailed allocation of the lesions to the respective risk categories including the absolute numbers (see table 2); we have moreover added the types of lesions in the “medium” and “high” risk group. Concerning your objection on a highly selected sample: Our study was performed in a clinical setting with primary aim to detect malignant lesions with a high sensitivity. We have therefore chosen this setting knowing that this is not fully applicable for the “general population”. We agree with you that this is a further limitation and have added this point in the limitations-section (see lines 202-205). 

The imaging procedure was easily done in most of the cases and usually no more than one attempt per lesion was needed in order to get a good image without any failure. The average imaging time including the recording of patient´s data was three minutes. We have added this information (see lines 134-137). 

Reviewer #2: 

The article is interesting, innovative and relevant, but some important information to analyze the data is missing.

I would suggest you make a Table with the frequency of images in each of the five categories and how many of them had the clinical evaluation or histopathological report. Also, it was not clear if you considered in your results all of the 238 patients or only those with biopsy or clinical evaluation?

Another question: did the dermatologists classified the images in low, medium and high risk or they were assigned automatically according to the diagnosis?

The agreement rate of diagnostic and risk classification could be stratified according to the lesions that were seen by dermatologists and had the histopathological report too. It is critical to know how the ground truth is being assigned to compare with the algorithm´s accuracy.

Reference 20 is equal to 22.

Authors response: We thank the reviewer for these comments. We have created a table based on your suggestions. All of the included 238 patients (irrespectively of having had a biopsy/excision) were considered for statistical analysis.

Yes, the dermatologists made a diagnosis and afterwards also classified the lesions in low, medium and high. 

We have deleted one reference and adapted the others accordingly. 

Editors’ comments:

"Michael Koppitz and Michael Tripolt share a company founded after finishing the study to produce a consumer-usable early skin cancer detection app. The remaining authors have no conflicts of interest to declare. "

Additional Editor Comments:

In accordance with the expert reviewers, I have a number of both technical and conceptual concerns. Rather than repeat those points here, I refer you to the specific remarks (below) for details. 

Authors response: We thank the Editor for these comments. 

1. We have revised our manuscript in order to meet the PLOS ONE's style requirements.

2. We have included the abovementioned Competing Interests statement to our updated cover letter.

3. We have addressed all your technical and conceptual concerns as also mentioned by the reviewers. 

Thank you for your help and re-consideration

Sincerely yours, 

Teresa Kränke, MD (corresponding author)

Michael Tripolt, MD, MPH 

Conflicts of interest: Michael Koppitz and Michael Tripolt share a company founded after finishing the study to produce a consumer-usable early skin cancer detection app. This does not alter our adherence to PLOS ONE policies on sharing data and materials. The remaining authors have no conflicts of interest to declare. 

Funding source: The development of the algorithms was funded by the amiflow Ltd., Graz, Austria; however, the authors did not receive any salaries by this company nor did it have any influence on the study design, data collection and analysis, decision to publish, or preparation of the manuscript.

---

## [Decision Letter · Decision Letter 1]

5 Aug 2022

PONE-D-22-08100R1New AI-algorithms on smartphones to detect skin cancer in a clinical setting – a validation studyPLOS ONE

Dear Dr. Kränke,

Thank you for submitting your manuscript to PLOS ONE. After careful consideration, we feel that it has merit but does not fully meet PLOS ONE’s publication criteria as it currently stands. Therefore, we invite you to submit a revised version of the manuscript that addresses the points raised during the review process.

I thank the authors for their revision. I agreement with the expert reviewers, there are still a number of concerns that need to be addressed. Please refer to the reviewers comments below.

We look forward to receiving your revised manuscript.

Kind regards,

Nikolas K. Haass, MD/PhD

Academic Editor

PLOS ONE

Journal Requirements:

Additional Editor Comments:

I thank the authors for their revision. I agreement with the expert reviewers, there are still a number of concerns that need to be addressed. Please refer to the reviewers comments below.

Reviewers' comments:

Reviewer's Responses to Questions

**Comments to the Author**

1. If the authors have adequately addressed your comments raised in a previous round of review and you feel that this manuscript is now acceptable for publication, you may indicate that here to bypass the “Comments to the Author” section, enter your conflict of interest statement in the “Confidential to Editor” section, and submit your "Accept" recommendation.

Reviewer #1: (No Response)

Reviewer #2: (No Response)

2. Is the manuscript technically sound, and do the data support the conclusions?

Reviewer #1: Partly

Reviewer #2: Yes

3. Has the statistical analysis been performed appropriately and rigorously? 

Reviewer #1: I Don't Know

Reviewer #2: Yes

4. Have the authors made all data underlying the findings in their manuscript fully available?

Reviewer #1: No

Reviewer #2: Yes

5. Is the manuscript presented in an intelligible fashion and written in standard English?

Reviewer #1: Yes

Reviewer #2: Yes

6. Review Comments to the Author

Reviewer #1: Thank you for the opportunity to review this revised submission. Although some changes have been made, I remain confused about several aspects of the study which I have documented below. One of the most important omissions is lack of data regarding lesion types, how the reference standard was reached for each group, and lack of data underlying the main accuracy results making it impossible to check the authors claims.

1. Additional information is provided about the two algorithms used, namely that they were previously developed and internally validated by the study authors and both are CE certified. The algorithms are identified as ‘analyze’ and ‘detect’ and seem to be incorporated into a smartphone app for skin cancer detection/risk assessment but the app itself is not identified. The supplementary information provides some information about the development and validation process however full details are not provided and this information does not appear to be in the public domain. As per my comment on the previous iteration of this paper, ideally one would one to see sufficient details of both the development and validation of the tools to allow the methods and results to be fully judged and to support any subsequent utilisation of these tools. As it stands it is not possible to associate these algorithms with any CE marked software or app.

2. Additional information is provided about methods for reaching a reference standard diagnosis for the included lesions, however the Results section does not report actual final diagnoses nor how they were reached (histology, clinical dx by two dermatologists or follow-up). It is also notable that the authors state that histology was the reference standard however only 165 lesions were reported as having a histological diagnosis. This is an important omission that needs to be addressed. It also appears that the clinical diagnosis was made in knowledge of the of algorithm recommendation (the last decision … was always made by the two dermatologists, even if the algorithm made an opposite risk classification). The lack of blinding to the algorithm recommendation must be clearly stated in the study report as, even with the best will in the world, knowledge of the index test result has clear potential to influence clinical decision making. The beginning of section ‘Procedure and Mobile Devices’ implies that clinical decision may have been made prior to use of the algorithm and could therefore have ensured some level of blinding – if this was the case then it should be stated more clearly.

3. Helpful clarification about the nature of the study population was provided. The recruitment process was still not described however (e.g. consecutive, convenience etc), and as per comment above, the actual final diagnoses of the included lesions were not reported.

4. I do not understand the authors’ meaning in the following sentence in their response (re lines 64-69) “Notably, no one, who did the index testing has screened the patients.”

5. I would still like to see a clearer separation of the description of the algorithms and their use from the reference standard diagnoses. I think that the algorithms provide both a diagnosis of lesion type and a risk classification but this has to be inferred from a number of different sections (line 58-59, 81-85, 87-96, 125-129) and could be more clearly documented.

6. The authors’ justification for excluding lesions rated as “benign” from calculation of specificity is inadequate. The paper also states that specificity was calculated both including and then excluding images of the risk category “benign response” but I cannot find this in the paper. An alternative approach would be to have two definitions of ‘index test positive’ for calculation of both sensitivity and specificity, ie. high risk (index positive) versus medium/low risk (index negative), and high or medium risk (index positive) versus low risk (index negative).

7. Table 2 provides the number of lesions per risk category as determined by each algorithm, and the number of lesions with a histological diagnosis, however these are the only absolute numbers provided. Ideally one would want to see a tabulation of the 47 lesion subcategories (or at least the five categories) against algorithm classification, i.e. number of lesions categorised as high/medium/low by the algorithm per lesion category. In addition, the absolute numbers underlying each percentage should be given e.g. for sensitivity 95.35%, the number of lesions classed as true positive and the total number of ‘malignant’ lesions should be reported. Thjs is a very important omission that must be addressed. It should also be made crystal clear that the accuracy data reported is for correct diagnosis of ‘possible malignancy’ as opposed to detection of already malignant skin lesions (i.e. includes detection of actinic keratosis or dysplastic nevus which are not universally considered even as ‘pre-malignant’).

8. Related to the above point, the Discussion mentions results that are not presented in the Results sections as far as I can see (i.e. lower ‘accuracy’ for detection of malignant lesions).

Reviewer #2: I found that the manuscript has gotten much better now after the review, but I still miss more data in the Results Section. I believe it is important to have a Table showing how many lesions were classified as benign and non-benign, including how many BCCs, SCCs, Melanomas, and how many of those had proven histopathological exam or only clinical diagnosis.

7. PLOS authors have the option to publish the peer review history of their article (what does this mean?). If published, this will include your full peer review and any attached files.

Reviewer #1: No

Reviewer #2: **Yes: **Mara Giavina-Bianchi

---

## [Author Response · Author response to Decision Letter 1]

18 Sep 2022

To the Editor

RE: Submission of the revised version of the manuscript number PONE-D-22-08100R1

Dear Editor, 

Thank you for giving us again the opportunity to re-submit our manuscript titled “New AI-algorithms on smartphones to detect skin cancer in a clinical setting”. Please find below the point-to-point answers to the suggestions made by the reviewers. 

Reviewer #1:

Thank you for the opportunity to review this revised submission. Although some changes have been made, I remain confused about several aspects of the study which I have documented below. One of the most important omissions is lack of data regarding lesion types, how the reference standard was reached for each group, and lack of data underlying the main accuracy results making it impossible to check the authors claims.

1. Additional information is provided about the two algorithms used, namely that they were previously developed and internally validated by the study authors and both are CE certified. The algorithms are identified as ‘analyze’ and ‘detect’ and seem to be incorporated into a smartphone app for skin cancer detection/risk assessment but the app itself is not identified. The supplementary information provides some information about the development and validation process however full details are not provided and this information does not appear to be in the public domain. As per my comment on the previous iteration of this paper, ideally one would one to see sufficient details of both the development and validation of the tools to allow the methods and results to be fully judged and to support any subsequent utilization of these tools. As it stands it is not possible to associate these algorithms with any CE marked software or app.

Author response: We thank the reviewer for these comments. Unfortunately, we are restricted to publicly share more data concerning the development and validation process of these both algorithms as this would infringe the copyright of both, the amiflow Ltd., Graz (funded the development) and the medaia Ltd., Graz (developer and owner of the App).

Moreover, as mentioned in the paper and the previous reply letter, we only performed a validation study of an already developed and CE-conform App. 

The developed App (Skinscreener©), however, is publicly available in the Google Play Store and the Apple Store. Please refer to: https://skinscreener.com/

2. Additional information is provided about methods for reaching a reference standard diagnosis for the included lesions, however the Results section does not report actual final diagnoses nor how they were reached (histology, clinical dx by two dermatologists or follow-up). It is also notable that the authors state that histology was the reference standard however only 165 lesions were reported as having a histological diagnosis. This is an important omission that needs to be addressed. It also appears that the clinical diagnosis was made in knowledge of the of algorithm recommendation (the last decision … was always made by the two dermatologists, even if the algorithm made an opposite risk classification). The lack of blinding to the algorithm recommendation must be clearly stated in the study report as, even with the best will in the world, knowledge of the index test result has clear potential to influence clinical decision making. The beginning of section ‘Procedure and Mobile Devices’ implies that clinical decision may have been made prior to use of the algorithm and could therefore have ensured some level of blinding – if this was the case then it should be stated more clearly.

Author response: We thank the reviewer for these comments. Concerning the final diagnoses of the lesions, we have added another table (table 3).

We stated the histology as ideal endpoint; in detail: Only those lesions, which were evaluated as being suspicious by the dermatologists, were removed – this procedure explains the low number of histologically proven lesions compared with the number of included lesions. We have adapted that paragraph (see lines 60-64). 

All lesions were first evaluated by two dermatologists and afterwards by the algorithms operated by a third dermatologist. Your concerns about the blinding and influence of the clinical decision are completely understandable, however, as the evaluating dermatologists had no idea about the algorithms´ evaluation the criteria for an adequately blinding-process are fulfilled in our view. According to your suggestion we have clarified the paragraph in the section “Procedure and Mobile Devices” (see lines 86-90).

3. Helpful clarification about the nature of the study population was provided. The recruitment process was still not described however (e.g., consecutive, convenience etc), and as per comment above, the actual final diagnoses of the included lesions were not reported.

Author response: We thank the reviewer for these comments. The recruitment process was consecutive; in detail: Every patient, who was seen by one of the study authors in the mentioned time period (June 2018-December 2019) was ask to participate in this study. We added this information (see lines 72-75). 

Concerning the final diagnoses please refer to table 3.

4. I do not understand the authors’ meaning in the following sentence in their response (re lines 64-69) “Notably, no one, who did the index testing has screened the patients.”

Author response: We thank the reviewer for this comment. This sentence indicates, that no author, who screened the respective patient did the screening with the smartphone. Please also refer to point 2 in this reply letter. 

5. I would still like to see a clearer separation of the description of the algorithms and their use from the reference standard diagnoses. I think that the algorithms provide both a diagnosis of lesion type and a risk classification but this has to be inferred from a number of different sections (line 58-59, 81-85, 87-96, 125-129) and could be more clearly documented.

Author response: We thank the reviewer for this comment. As you mentioned, these phrases were not clearly worded. The algorithms only provide a risk classification (benign/green, medium/yellow, high/red) of a respective lesion. In a previous training phase, both were trained with 18.384 lesions, which were allocated to one of the above-mentioned risk categories in each case. A list of lesions assigned to the risk groups “high” and “medium” is already given in the section “patients´ and lesions´ characteristics”. The lines you mentioned were reworded, wherever we considered useful.

6. The authors’ justification for excluding lesions rated as “benign” from calculation of specificity is inadequate. The paper also states that specificity was calculated both including and then excluding images of the risk category “benign response” but I cannot find this in the paper. An alternative approach would be to have two definitions of ‘index test positive’ for calculation of both sensitivity and specificity, i.e. high risk (index positive) versus medium/low risk (index negative), and high or medium risk (index positive) versus low risk (index negative).

Author response: We thank the reviewer for this comment. 

The results concerning the specificity (both, including and excluding the lesions of the risk category “benign” are mentioned in the section “Results/accuracy of the algorithms” (see lines 163-166). 

We agree with you that your approach would also be feasible; however, we mainly did the calculation as reported for the following reasons: In clinical practice the main goal is to identify possibly malignant, meaning suspicious lesions, as they require further medical action and/or treatment. Following this, it does not matter whether the suspicious lesions is allocated to the “yellow/medium” or the “red/high” risk group, as the respective lesion in any case has to be treated (i.e., follow-up in case of atypical nevi or histological examination in basal cell carcinoma). On the other hand, there is no need for action regarding lesions assigned to the “green/low” risk group. 

In our view, our approach corresponds in principle to your suggestion to calculate high or medium risk (index positive) versus low risk (index negative). However, as you suggested, we have calculated both, the sensitivity and specificity for “high and medium risk versus low risk” and added this information to the supplementary material. 

7. Table 2 provides the number of lesions per risk category as determined by each algorithm, and the number of lesions with a histological diagnosis, however these are the only absolute numbers provided. Ideally one would want to see a tabulation of the 47 lesion subcategories (or at least the five categories) against algorithm classification, i.e., number of lesions categorized as high/medium/low by the algorithm per lesion category. In addition, the absolute numbers underlying each percentage should be given e.g., for sensitivity 95.35%, the number of lesions classed as true positive and the total number of ‘malignant’ lesions should be reported. This is a very important omission that must be addressed. It should also be made crystal clear that the accuracy data reported is for correct diagnosis of ‘possible malignancy’ as opposed to detection of already malignant skin lesions (i.e., includes detection of actinic keratosis or dysplastic nevus which are not universally considered even as ‘pre-malignant’).

Author response: We thank the reviewer for these comments. Based on your suggestions we have added another table (table 3) and you will find there all missing information. 

8. Related to the above point, the Discussion mentions results that are not presented in the Results sections as far as I can see (i.e., lower ‘accuracy’ for detection of malignant lesions).

Author response: We thank the reviewer for this comment and totally agree with you that some data presented in the discussion were missing in the results. We have added that (see lines 166-168). 

Reviewer #2: I found that the manuscript has gotten much better now after the review, but I still miss more data in the Results Section. I believe it is important to have a Table showing how many lesions were classified as benign and non-benign, including how many BCCs, SCCs, Melanomas, and how many of those had proven histopathological exam or only clinical diagnosis.

Author response: We thank the reviewer for these comments. According to your suggestion we have added another table (table 3) containing all the missing information. 

Additional Editor Comments:

I thank the authors for their revision. I agreement with the expert reviewers, there are still a number of concerns that need to be addressed. Please refer to the reviewers’ comments below.

Author response: We thank the Editor for these comments. We have addressed all concerns made by the reviewers. 

We have created table 3 containing all missing information mentioned by the reviewers and added it to the main manuscript; however, this table could also be added to the supplementary material – whatever the Editor likes better. 

Thank you for your help and re-consideration

Sincerely yours, 

Teresa Kränke, MD (corresponding author)

Michael Tripolt, MD, MPH 

Conflicts of interest: Michael Koppitz and Michael Tripolt share a company founded after finishing the study to produce a consumer-usable early skin cancer detection app. This does not alter our adherence to PLOS ONE policies on sharing data and materials. The remaining authors have no conflicts of interest to declare. 

Funding source: The development of the algorithms was funded by the amiflow Ltd., Graz, Austria; however, the authors did not receive any salaries by this company nor did it have any influence on the study design, data collection and analysis, decision to publish, or preparation of the manuscript.

---

## [Decision Letter · Decision Letter 2]

11 Nov 2022

PONE-D-22-08100R2New AI-algorithms on smartphones to detect skin cancer in a clinical setting – a validation studyPLOS ONE

Dear Dr. Kränke,

Thank you for submitting your manuscript to PLOS ONE. After careful consideration, we feel that it has merit but does not fully meet PLOS ONE’s publication criteria as it currently stands. Therefore, we invite you to submit a revised version of the manuscript that addresses the points raised during the review process.

We look forward to receiving your revised manuscript.

Kind regards,

Mohamed Hammad, Ph.D.

Academic Editor

PLOS ONE

Journal Requirements:

Reviewers' comments:

Reviewer's Responses to Questions

**Comments to the Author**

1. If the authors have adequately addressed your comments raised in a previous round of review and you feel that this manuscript is now acceptable for publication, you may indicate that here to bypass the “Comments to the Author” section, enter your conflict of interest statement in the “Confidential to Editor” section, and submit your "Accept" recommendation.

Reviewer #1: (No Response)

Reviewer #2: (No Response)

2. Is the manuscript technically sound, and do the data support the conclusions?

Reviewer #1: Yes

Reviewer #2: Partly

3. Has the statistical analysis been performed appropriately and rigorously? 

Reviewer #1: Yes

Reviewer #2: Yes

4. Have the authors made all data underlying the findings in their manuscript fully available?

Reviewer #1: Yes

Reviewer #2: Yes

5. Is the manuscript presented in an intelligible fashion and written in standard English?

Reviewer #1: Yes

Reviewer #2: Yes

6. Review Comments to the Author

Reviewer #1: Thank you for the opportunity to review this revised submission. The paper is greatly improved and most of my earlier comments now addressed. The additional information provided regarding blinding, application of the reference standard and participant recruitment is very helpful.

1. I understand and accept that in this situation it is not possible to report information regarding earlier development and validation of the algorithms evaluated in this study, however it does not seem unreasonable to expect the authors to identify the algorithms’ association with Skinscreener app in the text of the paper, or is this also not possible?

2. Table 3 is an important and very useful addition to the main text. The link between the absolute numbers presented here and the percentage based results in the text is still rather opaque however, and could be much more clearly signposted, both by including absolute numbers underlying all %s reported in the text and including the % data in Table 3. For example, I believe it is possible to derive the reported sensitivities of the two algorithms (line 160), e.g. for analyze high or medium risk classification/total malignant/pre-malignant is (212+239)/473 = 95.35%, but it does not seem to be possible to similarly derive the numbers for the reported specificities. Similarly the data underlying the reported accuracies (76% for detect and 72% for analyze) ‘for malignant lesions compared to benign lesions’ cannot be derived. I believe that readers should not have to do so much work to try to understand the authors data and much more could be done to make it more accessible. Reporting of accuracy alone also makes it impossible to understand the source of and importance of the misclassifications – i.e. whether they are due to false negative or false positive results on the classifier. One does not want to risk missing malignant lesions however the potential health service implications from large numbers of false positive results cannot be under-estimated and must be taken into account.

3. As a final point, given the tertiary referral centre nature of the study population, I have some concerns about overstating of the implications of this study for use of these algorithms outside of a hospital setting, i.e. by patients themselves, without suitable caveats regarding potential for harm.

Reviewer #2: Dear authors,

Thank you for the chance to review your aticle, whicih is improving a lot. You have addressed partly what I had in mind in the last review, mainly including a Table 3 with some needed information.

I still think that is important for the analyses between the 2 softwares that you developed to include a table (it may be Table 3d) that shows histopathology against risk groups determined by the 2 softwares. After all, the preffered analyses was the histopathology and we don't see the results comparing the risk assesment by the 2 softwares versus the histopahology results.

Regarding the comment on line 166, we don't see any figure or table showing the results of this difference between malignant and non-malignant lesions. It would be nice to see a confusion matrix comparing malignant x non-malignant lesions for both softwares.

Best regards,

7. PLOS authors have the option to publish the peer review history of their article (what does this mean?). If published, this will include your full peer review and any attached files.

Reviewer #1: No

Reviewer #2: **Yes: **Mara Giavina Bianchi

---

## [Author Response · Author response to Decision Letter 2]

23 Dec 2022

To the Editor

RE: Submission of the revised version of the manuscript number PONE-D-22-08100R2

Dear Editor, 

Thank you for giving us again the opportunity to re-submit our manuscript titled “New AI-algorithms on smartphones to detect skin cancer in a clinical setting”. Please find below the point-to-point answers to the suggestions made by the reviewers. 

Reviewer #1:

Thank you for the opportunity to review this revised submission. The paper is greatly improved and most of my earlier comments now addressed. The additional information provided regarding blinding, application of the reference standard and participant recruitment is very helpful.

1. I understand and accept that in this situation it is not possible to report information regarding earlier development and validation of the algorithms evaluated in this study, however it does not seem unreasonable to expect the authors to identify the algorithms’ association with Skinscreener app in the text of the paper, or is this also not possible?

Author response: We thank the reviewer for this comment. The used AI in our study is the core of the App Skinscreener©, but not the App itself – in other words, the AI is the motor of the App, therefore we think it is not reasonable to identify the algorithms´ association with the App. Moreover, with the end of this year an updated version of the App will be available in the Google Play Store and the Apple Store. 

2. Table 3 is an important and very useful addition to the main text. The link between the absolute numbers presented here and the percentage-based results in the text is still rather opaque however, and could be much more clearly signposted, both by including absolute numbers underlying all %s reported in the text and including the % data in Table 3. For example, I believe it is possible to derive the reported sensitivities of the two algorithms (line 160), e.g., for analyze high or medium risk classification/total malignant/pre-malignant is (212+239)/473 = 95.35%, but it does not seem to be possible to similarly derive the numbers for the reported specificities. Similarly, the data underlying the reported accuracies (76% for detect and 72% for analyze) ‘for malignant lesions compared to benign lesions’ cannot be derived. I believe that readers should not have to do so much work to try to understand the authors data and much more could be done to make it more accessible. Reporting of accuracy alone also makes it impossible to understand the source of and importance of the misclassifications – i.e., whether they are due to false negative or false positive results on the classifier. One does not want to risk missing malignant lesions however the potential health service implications from large numbers of false positive results cannot be under-estimated and must be taken into account.

Author response: We thank the reviewer for these comments. 

As you suggested we have included the absolute numbers in the text and the underlying %s in the tables 3a-c (see lines 133-136 and updated tables 3a-c). 

Thank you for your concern about not to miss malignant lesions and the possible impact of a large number of false positive results on the health systems. Regarding that point, please refer to the study done by Tran et al. from 2005, who investigated the diagnostic accuracy of 272 general practitioners (GP) and six dermatologists in various skin conditions. Following their data, the overall diagnostic accuracy of the GPs was much lower than that of our investigated algorithms. 

Of course, no neural network will achieve a diagnostic accuracy of 100%, however, in comparison to abovementioned study, our neural networks surpassed the GPs; given that, our networks could have a positive impact on the health system as unnecessary visits and histological examination will be reduced. In this context, please also refer to a recent study done by Jinnai et al. 

We have added that point to our discussion (see lines 230-238) and both studies to our references (number 47 and 48). 

3. As a final point, given the tertiary referral center nature of the study population, I have some concerns about overstating of the implications of this study for use of these algorithms outside of a hospital setting, i.e., by patients themselves, without suitable caveats regarding potential for harm.

Author response: We thank the reviewer for this comment. You raise an important point as it is important to investigate, how valid such algorithms work outside a specialized hospital setting. For that reason, we have already performed a further study, in which exactly that issue (use of these algorithms in lay application) was examined. Preliminary results show that both sensitivity and specificity are lower as in this present study, however, are still above the reported ones of the general practitioners in the study mentioned at point 2 (see Tran et al. 2005). 

Reviewer #2: 

Dear authors, thank you for the chance to review your article, which is improving a lot. You have addressed partly what I had in mind in the last review, mainly including a Table 3 with some needed information.

I still think that is important for the analyses between the 2 softwares that you developed to include a table (it may be Table 3d) that shows histopathology against risk groups determined by the 2 softwares. After all, the preferred analyses were the histopathology and we don't see the results comparing the risk assessment by the 2 softwares versus the histopathology results.

Regarding the comment on line 166, we don't see any figure or table showing the results of this difference between malignant and non-malignant lesions. It would be nice to see a confusion matrix comparing malignant x non-malignant lesions for both softwares.

Best regards

Author response: We thank the reviewer for these comments. 

According to your suggestion we have added tables 4a and 4b, which show histopathology against the risk groups for both algorithms. 

Furthermore, we have created two crosstabulations (tables 3d and 3e) for both algorithms showing the algorithms´ risk * clinical category.

Thank you for your help and re-consideration

Sincerely yours, 

Teresa Kränke, MD (corresponding author)

Michael Tripolt, MD, MPH 

Conflicts of interest: Michael Koppitz and Michael Tripolt share a company founded after finishing the study to produce a consumer-usable early skin cancer detection app. This does not alter our adherence to PLOS ONE policies on sharing data and materials. The remaining authors have no conflicts of interest to declare. 

Funding source: The development of the algorithms was funded by the amiflow Ltd., Graz, Austria; however, the authors did not receive any salaries by this company nor did it have any influence on the study design, data collection and analysis, decision to publish, or preparation of the manuscript.

---

## [Decision Letter · Decision Letter 3]

6 Jan 2023

New AI-algorithms on smartphones to detect skin cancer in a clinical setting – a validation study

PONE-D-22-08100R3

Dear Dr. Kränke,

We’re pleased to inform you that your manuscript has been judged scientifically suitable for publication and will be formally accepted for publication once it meets all outstanding technical requirements.

Kind regards,

Mohamed Hammad, Ph.D.

Academic Editor

PLOS ONE

Additional Editor Comments (optional):

Reviewers' comments:

Reviewer's Responses to Questions

**Comments to the Author**

1. If the authors have adequately addressed your comments raised in a previous round of review and you feel that this manuscript is now acceptable for publication, you may indicate that here to bypass the “Comments to the Author” section, enter your conflict of interest statement in the “Confidential to Editor” section, and submit your "Accept" recommendation.

Reviewer #2: All comments have been addressed

2. Is the manuscript technically sound, and do the data support the conclusions?

Reviewer #2: Yes

3. Has the statistical analysis been performed appropriately and rigorously? 

Reviewer #2: Yes

4. Have the authors made all data underlying the findings in their manuscript fully available?

Reviewer #2: No

5. Is the manuscript presented in an intelligible fashion and written in standard English?

Reviewer #2: Yes

6. Review Comments to the Author

Reviewer #2: Dear authors,

The inclusion of the new table and detailed information about the lesions were very important for the compreheension of your article. Your paragraph with the 2 new references were also interesting.

Congratulations on your article! Best wishes,

Mara Giavina-Bianchi

7. PLOS authors have the option to publish the peer review history of their article (what does this mean?). If published, this will include your full peer review and any attached files.

Reviewer #2: **Yes: **Mara Giavina-Bianchi

---

## [Editor Report · Acceptance letter]

18 Jan 2023

PONE-D-22-08100R3 

New AI-algorithms on smartphones to detect skin cancer in a clinical setting – a validation study 

Dear Dr. Kränke:

I'm pleased to inform you that your manuscript has been deemed suitable for publication in PLOS ONE. Congratulations! Your manuscript is now with our production department. 

Kind regards, 

on behalf of

Dr. Mohamed Hammad 

Academic Editor

PLOS ONE